# The Hometown Is Hard to Leave, the Homesickness Is Unforgettable—The Influence of Homesickness Advertisement on Hometown Brand Citizenship Behavior of Consumers

**DOI:** 10.3390/bs13010054

**Published:** 2023-01-06

**Authors:** Chenwen Wei, Chao Wang, Lili Sun, Anxin Xu, Manhua Zheng

**Affiliations:** 1College of Economics and Management, Fujian Agriculture and Forestry University, Fuzhou 350002, China; 2College of Forestry, Fujian Agriculture and Forestry University, Fuzhou 350002, China

**Keywords:** homesickness advertising, hometown brand citizenship behavior, psychological ownership, place attachment

## Abstract

The hometown brand is symbolic of a hometown and can induce homesickness in consumers. Especially for those who live in other countries, it can stimulate their inner sense of belonging, self-identity, and effectiveness, in turn generating a series of brand behaviors and promoting sustainable brand development. In this study, we adopt a situational experimental research method from the perspective of sense of place theory and social exchange theory in order to assess the regulatory mechanism of place attachment through the mediating mechanism of psychological ownership. In this way, we explore the mechanism underlying the relationship between homesickness advertising (vs. non-homesickness advertising) on the hometown brand citizenship behavior of consumers. Our findings suggest that (1) homesickness advertising has a more positive effect on consumer hometown brand citizenship behavior than non-homesickness advertising; (2) psychological ownership plays a fully mediating role in the relationship between homesickness advertising (vs. non-homesickness advertising) and consumer hometown brand citizenship behavior; and (3) place attachment plays a moderating role in the influence of homesickness advertising (vs. non-homesickness advertising) on the hometown brand citizenship behavior of consumers through psychological ownership.

## 1. Introduction

In the current context of COVID-19 prevention and control, the new epidemic is still playing out, with ups and downs and small outbreaks all the time, making it increasingly difficult for people to return home every holiday season. In our modern society, with frequent population movements and increased life pressure, the outbreak of homesickness is an inevitable stage of social reform and development. People who are far away are haunted by the memories of their hometowns, and people under the high pressure of urban life also yearn for the traditional rural life, in order to return to the basics. Homesickness has a certain meaning, and has gradually become a social and cultural psychological phenomenon. There have been numerous classic advertising works related to homesickness. The various industries that play the homesickness card, whether it is agritourism, rural tourism, various consumer goods labeled to promote homesickness, or even homesickness advertising, entertainment programs, and so on, have been recognized and accepted by the public [1]. The most direct and common expression of homesickness theme in advertising is that people who are far away from their hometown miss their hometown and family during various festivals. From the early “Confucian house wine, making people homesick”, to the “Spring Festival Home” series of public service advertisements launched by CCTV during the Chinese New Year in recent years, such campaigns are all affectionate interpretations of homesickness advertising. At a time when remembering homesickness is strongly advocated, the hometown brand—as a symbol representing a specific region—can induce homesickness feelings in the hearts of consumers. Especially for those who live in other countries, adding the element of homesickness into hometown brand advertising can better stimulate an inner sense of belonging and self-identity. This will improve the hometown brand citizenship behavior of consumers and promote sustainable brand development.

According to previous studies, the view that consumers are “part-time employees” or “partial employees” of enterprises has been recognized by many scholars [2]. GROTH has formally proposed the concept of “customer citizenship behaviors (CCB)”, corresponding to the “part-time employee” status of consumers, on the basis of organizational citizenship behavior-related research. and defined it as a service that is not necessary for the successful production or delivery of a service. However, the spontaneous, haphazard behavior of consumers is generally good for the brand as a whole [3]. Hometown brands are those that originate from a consumer’s hometown. The civic behavior of consumers towards their hometown brands can help to reduce the operating costs of these brands. This allows the hometown brand to obtain more valuable business information and enhance their competitive advantage. The existing literature has mostly examined consumer brand behavior from the perspective of product functionality and rational cognition, which is considered to be weak [4,5,6,7]. Can homesickness advertising have an impact on consumers living in other countries, leading to a series of brand citizenship behaviors towards their home brands? What are the variables by which the mechanism of this role is realized? Can hometown branding play a role in the emotional content of the brand to promote sustainable development? These are the questions that must be answered, if such brands are to be further developed and broadened.

Using psychological ownership as a mediating variable and place attachment as a moderating variable, we investigate the effect of homesickness advertising on the hometown brand citizenship behavior of consumers, with the following main contributions:

First, we enrich the theoretical research related to brand citizenship behavior, thus providing a strong basis for sustainable development of consumer hometown brands. This study explores the relationships between homesickness advertising, psychological ownership, place attachment, and consumer brand citizenship behavior. The results suggest that consumer hometown brands can deliver value through consumer brand citizenship behaviors, and can play a role in promoting positive consumer purchases and even spreading positive word-of-mouth. This serves to promote the sustainable development of the branding path of hometown products.

Second, we provide a basis and guidance for corporate branding marketing, by exploring effective countermeasures for consumer brand behavior enhancement from the perspective of consumer psychological ownership. This allows brand production operators and relevant government departments to dig deep and cultivate such brands, by providing a theoretical basis and practical guidance to promote sustainable brand development.

The remainder of this paper is organized as follows: The theoretical background, conceptualization, and hypothesis development are presented first. Next, the research methodology, including the research design, study materials, and findings, is presented. The final section summarizes the analysis of the experimental findings, and suggests management insights to promote sustainable brand development.

## 2. Literature Review and Theoretical Hypotheses

### 2.1. Literature Review

#### 2.1.1. Homesickness Advertising

Homesickness advertising tends to focus on the homesickness memories of different consumer subjects. Homesickness can also be expressed in terms of folkloric events. To some extent, homesickness has developed accordingly with modernity. Liu has stated that, in advertising, the most common theme of homesickness is the longing for hometown and relatives, which is strongly reflected in the homesickness of people who are far away from home and work in different places. To a certain extent, this conforms to the interpretation of the meaning of “homesickness”, and also caters to the homesickness psychology of some advertising audiences [1]. He has analyzed 3 datasets including 4091 words from B station, revealing that some CCTV Spring Festival Gala PSAs have been broadcast for many years, but still continue to attract views and discussion on video sites [8]. Netizens have expressed their memories of the commercial and their emotions for their hometown through pop-ups. This also indicates that the CCTV Spring Festival Gala PSAs can invoke emotional resonance in viewers and summon people’s homesickness memories. Homesickness for one’s hometown is a permanent topic for Chinese people. Triggering emotional resonance in the advertising audience through the emotion of homesickness is an important path for traditional culture, in order to improve the psychological fit of audiences in advertising media. The creation of homesickness and re-interpretation of folklore imagery cater to the homesickness mentality of the audience, mobilizing people’s memories of the past, providing a convenient means for the development of folklore resources, and helping to promote the regeneration and recognition of traditional folklore [9].

#### 2.1.2. Consumer Hometown Brand Citizenship Behavior

The concept of consumer brand citizenship behavior originates from the research field of Organizational Citizenship Behavior. Gruen was the first to apply the concept of “citizen behavior” in organizational behavior to research consumer behavior in marketing [10]. Groth formally defined consumer brand citizenship behavior as follows: consumers voluntarily adopt products or services that benefit brands. However, it is not a necessary measure for brands to provide products and services. Voluntary initiative, positivity, and non-role are important characteristics of consumer brand citizenship behavior [3]. Chen has defined consumer brand citizenship behavior as a spontaneous non-consumption behavior or service production behavior outside the consumer’s role, which is generally beneficial to the brand or other consumers [11]. Consumer hometown brand citizenship behavior is a new concept in the field of consumer behavior research, which means that consumers voluntarily present valuable and constructive behaviors for hometown brands outside their roles. This can lead to low-cost (or even no-cost) competitive advantages for hometown enterprises. For example, consumers spontaneously recommend, through word-of-mouth behavior, their hometown brand to their friends; are willing to show that they own the products or services of the hometown brand for publicity; actively cooperate with the brand’s return visit research activities or new product development activities; and are willing to cooperate with the brand wait. Li has stated that regional brands clearly and specifically inform the geographical origin of products [12]. Therefore, they can activate the sense of self-identity in consumers from the same region, providing important homesickness sustenance for people who are far away from their hometown and miss their hometown in the modern context of rapid urbanization and modernization. Zhang has pointed out that consumers are also more inclined to accept marketing strategies that conform to their self-identity. Therefore, people tend to support brands from their hometown in order to gain an emotional sense of belonging. When a person’s homesickness is relatively high, they are more likely to transfer their homesickness and love for their hometown to their hometown brands [13].

#### 2.1.3. The Correlation between Psychological Ownership and Consumer Perception and Behavior

In recent years, academics have conducted exhaustive research on psychological ownership, the scenarios that generate it, and the mechanisms that constitute it. Pierce has argued that psychological ownership can arise when the goal satisfies different claims within the individual [14]. The core of psychological ownership is the feeling of “possession” of the target, rather than the actual right to it. The root of the creation of psychological ownership is the psychological connection between the individual and the goal [15]. It can be generated due to emotional energy. For example, the trial experience before buying a product can creates emotions such as “love” and “trust” for the brand. Gawronski has argued that psychological ownership can arise through individual subjective cognitive feelings, such as determining the scarcity of a product before purchase, feeling the unique value of the product, and so on [16]. Psychological ownership has been shown to be promoted in consumers prior to purchasing hometown brand agricultural products [17]. The necessary condition for its creation is the degree of effort that the individual continuously devotes to the target object, such as energy and time. There are four main motivations: a sense of psychological belonging, a sense of efficacy, a sense of self-identity, and a sense of autonomous control. Li has stated that these four motives follow three pathways to generate psychological ownership. The most central way is to control an object, as well as to understand an object deeply, or to transform the self into an object or become psychologically associated with an object [18].

#### 2.1.4. Place Attachment

Williams and Roggenbuck first proposed the concept of “Place Attachment”, that is, an individual’s sense of belonging to a special place [19]. In the field of brand marketing, scholars have extended the attachment relationship in psychology from “people and people” to “people and things”. The higher the degree of emotional association between the consumer and a place, the higher the consumer’s preference for local brand products, resulting in loyal buying behavior, ultimately achieving corporate brand marketing goals. In terms of commonality, scholars believe that the emotional connection of consumers to places is an important part of place attachment, and have emphasized the important role of emotional investment. Takamatsu has stated that place attachment refers to a unique emotional bond between an individual and a place, and the strength of this relationship is important to whether the individual wants to spend more time, energy, and money on the place [20]. Stefaniak et al. have further defined it as a positive emotional connection between an individual and a specific place, with the main characteristic being that the individual tends to maintain a long-term and close connection with the place [21]. In terms of personality, the concept of place attachment by different scholars includes a personalized part, focused on bond relationship and emphasized feelings, among other aspects [22]. 

From the above-mentioned combinations of related theories, domestic and foreign scholars have adopted different theories and approaches. The influencing factors of brand citizenship behavior have been studied from various perspectives, and rich theoretical research results have been achieved. However, consumer brand citizenship behavior is still a relatively new area. Foreign research on consumer brand citizenship behavior is only a decade old, and no scholar has studied the relationship between homesickness advertising and consumer brand citizenship behavior. Therefore, in this study, we attempt to explore the mechanism of the hometown brand homesickness advertising orientation on the brand citizenship behavior of consumers from the perspective of psychological ownership.

### 2.2. Research Hypothesis and Model Design

#### 2.2.1. Homesickness Advertising (vs. Non-Homesickness Advertising) and Hometown Brand Citizenship Behavior

Under modern emotional consumerism, goods are not only a simple object system, but also a symbolic system with both use value and emotional value. In the process of advertising communication, the content itself carries a certain symbolic and representative meaning. In the case of the psychological fit between the advertising audience and the advertisement, this symbolic and representative meaning also derives from the audience’s psychological judgment of the value of the advertising content [23]. Homesickness for one’s hometown is a permanent topic, and the emotion of homesickness triggers resonance in the advertising audience. It is also an important path for traditional culture to improve the psychological fit of audiences in advertising media. For example, the classic Chinese advertisement “Confucius Family Wine, makes you homesick” combines homesickness with wine, suggesting that drinking Confucius Family Wine provides a way to relieve homesickness to the audience. In this way, the sense of fit and identity between the audience and the advertisement is increased, based on the sense of place theory, as a commodity with both use value and emotional value. When consumers watch homesickness advertising, as hometown brands carry hometown culture, they are more likely to serve as an emotional support and generate a sense of belonging. After obtaining a certain level of emotional satisfaction, consumers will engage in a series of active brand citizenship behaviors, based on the social exchange theory and the principle of reciprocity [24,25].

In summary, the homesickness of consumers can be relieved when they obtain information about products from their hometown. The consumers will, thus, spontaneously generate positive behaviors in favor of their hometown brands. Therefore, we propose the following hypothesis:

**H1.** *The use of homesickness advertising (vs. non-homesickness advertising) is more likely to elicit consumer behavior toward hometown brand citizenship*.

#### 2.2.2. Homesickness Advertising (vs. Non-Homesickness Advertising) and Consumer Hometown Brand Citizenship Behavior: The Mediating Role of Psychological Ownership

According to the sense of place theory, the hometown carries the hometown feeling of consumers; therefore, the hometown brand can promote a sense of belonging in consumers. From the perspective of belonging, the land, family, and culture of the hometown create strong emotions in people, as they have a sense of ownership of their hometown (e.g., “mine” and “ours”). From the perspective of self-association, the concept of hometown, where geo-organization and familial organization overlap, creates a strong bond. Individuals are also likely to form psychological ownership of their hometown brand, based on the homesickness brought about by hometown complexes and a sense of belonging. From an emotional point of view, people’s innermost memories create homesickness for their hometowns, for the places they have lived or experienced, and familiar things. In this way, the most affectionate and beautiful emotion for the hometown arises [26].

Consumer place awareness, hometown feelings, and sense of belonging allow individuals to create a close bond with their hometown brand and self. At the same time, consumers have a close relationship with their hometown brands, and the constant awareness of such brands makes them more familiar with it, therefore creating psychological ownership [27]. By exploring the roots of psychological ownership, hometown brands enable consumers to feel a sense of ownership, self-identity, and emotion, creating psychological ownership [28]. According to social exchange theory, consumers who develop psychological ownership of their hometown brand will enhance their brand behavior, as a result. This is demonstrated by buying more, paying a premium, word-of-mouth referrals, and so on, as well as always being ready to maintain the brand. In the stimulus–individual physiological/psychological response pathway, utilization of the homesickness advertising context induces and changes the emotional experience of consumers. This affects emotional changes in consumers and creates psychological ownership of the hometown brand. After a series of physiological and psychological reaction process, consumers will enact a behavioral response to the stimulus, such as re-purchasing, recommendation, and other convergent behavior brand citizenship behavior.

In summary, when companies highlight homesickness emotions in their advertising content, it will make consumers feel homesick. This causes the individual to create a strong bond between the hometown brand with the self, thus generating psychological ownership of the hometown brand, which triggers brand citizenship behavior. Therefore, we propose the following hypothesis:

**H2.** *Psychological ownership plays a mediating role in the relationship between homesickness advertising (vs. non-homesickness advertising) and consumer brand citizenship behavior, and homesickness advertising (vs. non-homesickness advertising) is more likely to elicit psychological ownership of hometown brands and promote the formation of brand citizenship behavior in consumers*.

#### 2.2.3. The Moderating Effect of Place Attachment

From a psychological perspective, Chen has argued that an individual’s outward attachment can be motivated by place-shaping behaviors, such as conscious marketing [25]. Implicit attachment, on the other hand, is formed after an individual’s personal experience at a destination and after interacting with the place, following which epiphenomenal attachment is transformed into implicit attachment. Consumer emotional attachment is the cognitive, emotional, and intentional psychological bond that connects consumers to a specific consumer object, comprising an emotional connection based on a holistic experience. It can trigger a strong motivation for consumers to invest their own resources to maintain the relationship; that is, emotional attachment produces the specific behaviors needed to enhance the relationship, and strong motivation and behavioral tendencies are the most important attributes of emotional attachment. Specifically, a consumer’s attachment to the place of production of their hometown brand will shape their sense of belonging to and self-association with that brand. This, in turn, increases the consumer’s psychological ownership of the brand. When consumers establish a brand identity relationship, it will prompt them to create an emotional connection with the brand and care about it deeply. This leads to more voluntary extra-role behaviors that benefit the brand. According to social exchange theory, in order to maintain the relationship, individuals will make a return on a relationship that has gained benefits. This drives reciprocal exchange behavior and directly contributes to the construction and maintenance of individual identity and attachment relationships. Consumers with strong emotional attachment are more eager to invest a lot of time, effort, and money into a brand or company in the area they are attached to. This also stimulates higher-level behavioral responses, such as spreading positive word-of-mouth to relatives and friends, enthusiastically joining the company’s brand community, reducing price sensitivity, or even preferring to wait for products out of loyalty when out of stock [29]. When consumers establish an emotional attachment connection with a brand, they are comforted and reassured, due to their psychological well-being. This affects their perception of the brand as the only choice and leads to engagement in word-of-mouth communication, product recommendation, and other product support behaviors.

In summary, the degree of emotional place attachment to their hometown is what creates psychological ownership of hometown brands in consumers. This, in turn, affects the citizenship behavior of consumers toward their hometown brands. Therefore, the following hypothesis is proposed:

**H3.** *High place attachment reinforces the mediating effect of homesickness advertising (vs. non-homesickness advertising) through psychological ownership of consumer brand citizenship behaviors. That is, the mediating effects described above have a higher degree of impact on consumers high place attachment than those with low place attachment*.

#### 2.2.4. Research Model Construction

This study is based on the question of whether homesickness advertising can elicit citizenship behavior in consumers relating to their hometown brands in practice. Based on a review of the relevant literature, consumer hometown brand advertising can be divided into homesickness advertising and non-homesickness advertising. To investigate the mechanism of its influence on the citizenship behavior of consumers, with respect to their hometown product brands, and to investigate the direct influence of homesickness advertising on the citizenship behavior of consumers towards hometown brands, the mediating role of psychological ownership was extrapolated in relation to the moderating role of place attachment. A total of three research hypotheses were developed. Based on the above research relationship assumptions, the theoretical model framework of this study is shown in Figure 1 below.

## 3. Experimental Design and Analysis of Results

### 3.1. Pre-Experiment

#### 3.1.1. Experimental Purpose

To ensure that homesickness advertisements are manipulated in such a way that they effectively stimulate homesickness feelings in consumers for their hometowns, the variables involved in the main experiment (homesickness ads vs. non-homesickness ads) were tested against each other before the formal experiment.

#### 3.1.2. Experimental Design and Subjects

We assessed representations of corporate marketing advertising in practice and experimental materials from the literature on advertising-oriented related research [30,31]. This study refers to the real advertising slogans of a Chinese brand. After the expert panel discussion, two sets of advertisements with posters and textual representations were proposed as stimulus materials. The materials were selected from advertisements of similar products on Taobao, a Chinese e-commerce platform. One group emphasizes the emotional attributes of homesickness, while the other emphasizes the functional attributes of the product. To avoid the interference of irrelevant factors, the word count of the two advertisements was kept roughly the same. In addition, to avoid bias caused by past experiences and individual differences of consumers, virtual brands were used in this study.

The pre-experimental subjects were mainly obtained using a random sampling principle. A total of 32 subjects were recruited and asked to read each of the two advertisements. They were then asked to score the items measured, in order to test whether the poster and text content were effectively perceived by consumers as homesickness advertising versus non-homesickness advertising. After the subjects read the relevant materials, they were asked to respond to the question “What is the type of the above advertisement?” (rated from 1 = “homesickness advertising” to 7 = “non-homesickness advertising”). None of the 32 subjects who participated in the pre-experiment took part in the formal experiment.

#### 3.1.3. Analysis of Experimental Results

For the manipulation test of the experimental material of advertising claims, 32 subjects (14 males, 18 females; age, 20–60 years old) were randomly selected and equally assigned to two experimental situations for the manipulation test. Web-based questionnaires were distributed through the Credamo platform, and data were processed using the SPSS 26.0 software. Invalid data with the same answers were removed, leaving 32 valid data with an efficiency rate of 100%. The experimental results indicated that the homesickness emotion dimension scores were significantly higher in the homesickness category advertisement group than in the non-homesickness group [M_homesickness_ = 5.44, SD = 1.153; M_non-homesickness_ = 2.06, SD = 0.680; t(32) = 10.086, *p* < 0. 005]. The experimental results demonstrated that, in the sub-group shown non-homesickness advertisements, individuals perceived stronger functional characteristics of the hometown brands than homesickness perceptions; meanwhile, the homesickness emotions felt by individuals in the homesickness category ad group were stronger than the functional properties of the product. The experimental material manipulation test was successful and the two ads were applied to all of the following experiments.

### 3.2. Experiment 1: The Effect of Homesickness Advertising (vs. Non-Homesickness Advertising) on Hometown Brand Citizenship Behavior of Consumers

#### 3.2.1. Pre-Experiment

The purpose of the pre-experiment was to test the validity of the experimental stimulus material reflecting the experimental manipulated material (i.e., homesickness ads vs. non-homesickness ads). For this study, the situational simulation experimental method was used. Tea leaves were used as the experimental stimulus material. The ad was designed in the form of “picture + text”. The material was selected from the advertisements of similar products on Taobao, a Chinese e-commerce platform.

The pre-experiment was conducted in a random sample of 30 subjects (14 female, 16 male; age, 20–50 years) through Credamo’s sample recommendation service, in order to ensure a balanced number of subjects under different stimulus conditions. Furthermore, to avoid interference from subject demographic variables, the questionnaires were distributed randomly in both groups. We also made sure that each participant read and answered only one of the advertising slogans and related questionnaires. First, the same introductory scenario was presented to all subjects: Suppose you see a tea from your hometown for sale. The subjects then randomly viewed one of the two sets of material presented. The first group read promotional materials with text and pictures characterized by homesickness advertising. The advertising material was designed with reference to the homesickness commercials broadcast during the Chinese New Year. The second group read non-homesickness advertising, mainly describing the efficacy of tea. The subjects rated the hometown brand promotion advertisement on a semantic difference scale after reading the above material (from 1 = homesickness advertisement to 7 = non-homesickness advertisement).

The results of the independent samples *t*-test indicated a significant difference between the mean scores of the first and second groups [M_homesickness_ = 1.60, SD = 1.056 vs. M_non-homesickness_ = 3.67, SD = 1.988; t(15) = 3.556, *p* < 0.001]. Therefore, the group of stimulus materials was tentatively considered valid. The effect of homesickness advertising was further tested using analysis of covariance (ANCOVA), controlling for the age and gender of the subjects. The results were consistent with the one-way ANOVA. The specific results indicated no significant effect of gender on homesickness advertising vs. non-homesickness advertising [F(1,27) = 0.66, *p* = 0.800], and no significant effect of age on homesickness advertising vs. homesickness advertising [F(1,27) = 0.424, *p* = 0.516]. Similar pre-experiments were conducted for all the following experiments but, for the sake of brevity, this step of the analysis is not presented (see Results section).

#### 3.2.2. Formal Experiments

The purpose of Study 1 was to test the effect of homesickness advertising (vs. non-homesickness advertising) on the hometown brand citizenship behavior of consumers (i.e., to test Hypothesis H1). Therefore, a one-way two-level (homesickness advertisements vs. non- homesickness advertisements) between-group experiment was designed. For the formal experiment, we recruited 160 subjects (77 females, 83 males; age, 20–60 years) on the Seeing Numbers platform. Subjects were randomly assigned to the symbolic homesickness advertising versus the non-homesickness advertising. Both groups of subjects read the experimental material in the pre-experiment and filled in the consumer hometown brand citizenship behavior questions, as well as demographic information, after they finished reading. The Consumer Hometown Brand Citizenship Behavior Scale was adapted from Becerra et al. [28], including a scale with 4 questions. For example, “If people around me need to buy similar products, I would recommend them to buy the brand advertised above in preference to other hometown brands” and “If the brand is conducting a consumer survey, I would actively participate in it compared to other brands and provide suggestions for improvement.” The question items used a 7-point Likert scale, ranging from 1 = strongly disagree to 7 = strongly agree. After eliminating 6 invalid questionnaires, 77 valid questionnaires were obtained for the homesickness advertising group vs. 77 for the non-homesickness advertising group.

#### 3.2.3. Analysis of Results

First, the Cronbach’s coefficient of the consumer hometown brand citizenship behavior test was 0.841, which is greater than 0.7; therefore, the scale passed the reliability test. Leven’s test showed that F(1,152) = 0.007 and *p* = 0.936, allowing for further one-way ANOVA analysis (the experimental analysis of the study series passed this test and, for brevity, the following experimental results are omitted to simply report the data results for this step). Second, the results of the one-way ANOVA test showed a significant difference in the effect of homesickness advertising (vs. non-homesickness advertising) on consumer hometown brand citizenship behavior [F(1,156) = 4.796, *p* < 0.005]. The results of the independent samples *t*-test also indicated that the homesickness advertising group was more likely to enhance consumer hometown brand citizenship behavior than the non-homesickness advertising group [M_homesickness_ = 5.243, SD = 1.113 vs. M_non-homesickness_ = 4.866, SD = 1.018; t(152) = 2.190, *p* = 0.030]; see Table 1.

In summary, Study 1 provided preliminary evidence that homesickness advertising significantly enhances consumer brand citizenship behavior, compared to non-homesickness advertising. As a result, hypothesis H1 was validated. Tea was chosen as the experimental stimulus in Study 1. In Experiment 2, the experimental material was replaced to form new stimulus material, in order to extend the external validity of the study model. In particular, the mediating effects of collective efficacy and risk perception were further explored.

### 3.3. Experiment 2: The Mediating Role of Psychological Ownership

#### 3.3.1. Pre-Experiment

The pre-experiment for Study 2 replicated the advertising approach of Study 1, but replaced the experimental material, tea, with passionfruit. In this study, 30 subjects (14 female, 16 male; age, 20–60 years) were randomly selected through the sample recommendation service of Credamo. The results showed that there was a significant difference between the subject’s material ratings for homesickness advertising and non-homesickness advertising [M_homesickness_ = 1.21, SD = 0.426 vs. M_non-homesickness_ = 5.20, SD = 1.859; t(18) = 8.078, *p* < 0.001]. Thus, the stimulus material in this group proved to be effective.

#### 3.3.2. Formal Experiments

The purpose of Study 2 was, first, to revalidate the results of Study 1 and enhance the external validity of the theoretical model and, second, to test the mediating effect of psychological ownership (i.e., to test Hypothesis H2). A one-way two-level between-groups experiment (homesickness advertising vs. non-homesickness advertising) was designed. For the formal experiment, we recruited 180 subjects (106 females, 174 males; age, 20–60 years) in the Seeing Numbers platform. Subjects were randomly assigned to homesickness ads (vs. non- homesickness ads), and read the experimental materials in the pre-experiment separately. After reading the materials, subjects were required to fill in the consumer hometown brand citizenship behavior questions, psychological ownership questions, and demographic characteristics in turn. The consumer hometown brand citizenship behavior scale was the same as in Experiment 1, and the psychological ownership scale was adapted from the Van Dyne and Pierce scale with 5 questions [32,33], including “I feel that the brand belongs to my hometown”, “ I feel that the brand has a strong connection with me”, “I have a strong affinity for the brand”, and so on. All scale items were on a 7-point Likert scale, ranging from 1 = strongly disagree to 7 = strongly agree. After excluding 20 invalid questionnaires, 80 valid questionnaires were obtained for the homesickness advertising group and 80 for the non-homesickness advertising group.

#### 3.3.3. Analysis of Results

First, reliability tests were conducted. The results showed that the Cronbach’s coefficient of consumer hometown brand citizenship behavior was 0.790 and the Cronbach’s coefficient of psychological ownership was 0.915, both of which were greater than 0.7; therefore, the scales passed the reliability test.

Second, an independent samples *t*-test was conducted to verify the main effect of homesickness advertising on the citizenship behavior toward hometown brands by consumers. The results showed that consumers showed higher civic behavior towards hometown brands under homesickness advertisements, compared to non-homesickness advertisements [Mh_omesickness_ = 5.534, SD = 0.814 vs. M_non-homesickness_ = 4.469, SD = 1.166; t(141) = −6.074, *p* < 0.001], demonstrating a significant main effect. The same independent samples *t*-test was used to test the effect of gender on mediating variables. The results showed that consumers showed stronger psychological ownership of the hometown brand with the homesickness advertising [M_homesickness_ = 5.405, SD = 0.892 vs. M_non-homesickness_ = 4.005, SD = 1.412; t(133) = −7.500, *p* < 0.001]; see Table 2.

Finally, the mediating effect of psychological ownership was tested. Using the SPSS26.0 statistical software process4.1 plug-in, the Bootstrap sampling number was set to 5000, the confidence interval was 95%, and model 4 was selected [34]. The results indicated that the direct effect of homesickness advertising (vs. non-homesickness advertising) on consumer hometown brand citizenship behavior was not significant (β_direct effect_ = 0.199, 95% CI = [−0.082, 0.480], including 0); meanwhile, the indirect effect was significant (β_indirect effect_ = 0.766, 95% CI = [0.524, 1.046], excluding 0). Specifically, the mediating effect of psychological ownership was significant (β_indirect effect_ = 0.766, 95% CI = [0.524, 1.046], excluding 0); see Table 3 and Table 4 and Figure 2. Thus, psychological ownership mediated the effect between homesickness advertising (vs. non-homesickness advertising) and consumer hometown brand citizenship behavior, and hypothesis H2 was supported.

The results of Studies 1 and 2 indicated that homesickness advertising has a more positive impact on consumer hometown brand citizenship behavior, compared to non- homesickness advertising. In addition, homesickness advertising (vs. homesickness advertising) influenced consumer hometown brand citizenship behavior through psychological ownership. The first two studies briefly described homesickness advertising as a means to explore the effect of homesickness advertising (vs. non-homesickness advertising) on consumer hometown brand citizenship behavior. In Study 3, we examined how consumer hometown brand citizenship behavior changes in response to changes in consumer place attachment, in order to explore the boundary conditions of the influence of homesickness advertising (vs. non-homesickness advertising) on consumer hometown brand citizenship behaviors.

### 3.4. Experiment 3: The Moderating Effect of Place Attachment

#### 3.4.1. Pre-Experiment

The purpose of the pre-experiment was to test the validity of consumer place attachment (high vs. low) and the experimental material. The Likert 7-point scale was used to measure the place attachment of subjects, with reference to the scale of Williams [19]. We divided the place attachment into two groups—high vs. low—using a value of 4 as the cut-off point. In this study, a random sample of 62 subjects (38 female, 24 male; age, 20–60 years) was surveyed through the sample recommendation service of the Seeing Numbers platform. First, the subjects were asked to fill out a place attachment scale, which divided them into high and low attachment groups. The pre-experiment of Study 3 replicated the advertising campaign of Study 2, but replaced the experimental material, passionfruit, with peanuts. The results of the experiment indicated that there was a significant difference in the responses of subjects to homesickness advertisements vs. non-homesickness advertisements [M_homesickness_ = 1.68, SD = 0.871 vs. M_non-homesickness_ = 4.33, SD = 2.040; t(35) = 6.575, *p* < 0.001]. Therefore, the homesickness advertising (vs. non-homesickness advertising (i.e., the independent variable) stimulus material was considered to be effective.

#### 3.4.2. Formal Experiments

The purpose of Study 3 was to test the moderating role of place attachment, based on the first two studies; that is, how the role of place attachment moderates between homesickness advertising (vs. non-homesickness advertising), psychological ownership, and consumer hometown brand citizenship behavior. A 2 (homesickness advertising vs. non-homesickness advertising) × 2 (place attachment: high vs. low) between-group experimental design was used. The formal experiment was conducted on the Seeing Numbers platform, and a total of 360 subjects (184 females, 176 males; age, 20–60 years) were recruited. The subjects were also randomly assigned to one of four experimental scenario groups: homesickness advertising × high place attachment, homesickness advertising × low place attachment, non-homesickness advertising × high place attachment, or non-homesickness advertising × low place attachment.

First, the subjects read the advertising materials used for the experiment. After the four groups of subjects read the corresponding advertising messages, they were asked to fill in the same consumer hometown brand citizenship behavior questions and psychological ownership questions as in Experiment 2, with all questions on a 7-point Likert scale ranging from 1 = strongly disagree to 7 = strongly agree. Finally, they filled in demographic information. After eliminating 72 invalid questionnaires, there were 71 questionnaires in each of the 4 groups, for a total of 284 questionnaires.

#### 3.4.3. Analysis of Results

First, reliability tests were conducted. The results showed that the Cronbach’s coefficient of the consumer hometown brand citizenship behavior scale was 0.809, the Cronbach’s coefficient of the psychological ownership scale was 0.915, and the Cronbach’s coefficient of the place attachment scale was 0.816, all of which were greater than 0.7. Therefore, the scales passed the reliability test.

Second, we carried out an independent samples *t*-test. The results showed that the effect of homesickness advertising (vs. non-homesickness advertising) on consumer hometown brand citizenship behavior [M_homesickness_ = 5.444, SD = 0.833 vs. M_non-homesickness_ = 4.684, SD = 1.109; t(265) = −6.574, *p* < 0.001] and psychological ownership [M_homesickness_ = 5.422, SD = 1.267 vs. M_non-homesickness_ = 4.079, SD = 1.267; t(266) = −10.147, *p* < 0.001] had significant differences in impact; that is, homesickness advertisements were more likely to stimulate brand citizenship behavior and psychological ownership of hometown brands in consumers than non-homesickness advertisements. Thus, hypotheses H1 and H2 were validated.

Next, the moderating effects of place attachment on homesickness advertising (vs. homesickness advertising), psychological ownership, and consumer hometown brand citizenship behavior were examined. The results of a two-factor analysis of variance (MANOVA) revealed that the interaction term between place attachment and homesickness advertising (vs. non- homesickness advertising) had a significant effect on psychological ownership [F(3,284) = 62.933, *p* < 0.001].

To test the mediating effect of being moderated, we referred to the method proposed by Edwards and Lambert [35], and assessed the size of the mediating effect and whether there was a difference in significance in order to determine whether the mediating strength is moderated at high and low levels of the moderating effect. Using the PROCESS 4.1 plug-in for SPSS 26.0 statistical software, the Bootstrap sampling number was set to 5000, the confidence interval was 95%, and model 7 was selected [30].

The results indicated (see Table 5) that the direct effects of homesickness advertising (vs. non-homesickness advertising) and consumer hometown brand citizenship behavior did not reach significant levels when psychological ownership was not included (β_direct effect_ = −0.007, LLCI = −0.208, ULCI = 0.193, interval with 0). Under high place attachment, homesickness advertising (vs. non-homesickness advertising)→psychological ownership→consumer hometown brand citizenship behavior (β_indirect effect_ = 0.952, LLCI = 0.710, ULCI = 1.214, interval not including 0) reached a significant level; meanwhile, under low place attachment, homesickness advertising (vs. non-homesickness advertising)→psychological ownership→consumer hometown brand citizenship behavior (β_indirect effect_ = 0.582, LLCI = 0.372, ULCI = 0.832, interval without 0) before the mediating effect reached a significant level. Thus, Study 3 further verified the boundary conditions for the existence of homesickness advertising through the mediating effect of place attachment (high vs. low).

## 4. Conclusions and Recommendations

### 4.1. Research Summary

In this study, we used an experimental research method from the perspective of sense of place theory and social exchange theory in order to demonstrate the effects of homesickness advertising (vs. homesickness advertising) on the hometown brand citizenship behavior of consumers. Psychological ownership was introduced to explain the mediating mechanisms involved, and the moderating role of place attachment in the above effects was proposed and tested. The results of the study demonstrated that ① homesickness advertisements were more likely to stimulate citizenship behavior toward hometown brands in consumers than non-homesickness advertisements (Experiment 1). Specifically, when consumers watch homesickness advertising, as hometown brands carry hometown culture, they are more likely to provide emotional support and generate a sense of belonging. Consumers receive a certain level of emotional satisfaction and, out of reciprocity, develop a series of active brand citizenship behaviors. ② Psychological ownership fully mediates the relationship between homesickness advertising (vs. non-homesickness advertising) and consumer hometown brand-citizenship behavior (Experiment 2). Specifically, homesickness advertising can influence emotional changes in consumers, creating psychological ownership of hometown brands. After a series of physiological and psychological reactions, consumers will present certain behavioral responses to the stimulus, such as re-purchasing, recommendation, and other convergent brand citizenship behaviors. ③ Homesickness advertising (vs. non-homesickness advertising) plays a moderating role in the effect of place attachment on the hometown brand citizenship behavior of consumers through psychological ownership (Experiment 3). Specifically, attachment to the place of production of their hometown brand will shape a consumer’s sense of belonging and self-efficacy to that brand. This, in turn, increases the consumer’s psychological ownership of the brand. When consumers establish a brand identity relationship, they will create an emotional connection with the brand and deeply care about it. This serves to stimulate more voluntary extra-role behaviors, which benefit the brand and promote sustainable brand development.

### 4.2. Research and Discussion

First, domestic and foreign scholars have used different theories and methods to study consumer brand behaviors from the perspectives of perceptions and payments, purchase intentions [12], and preferences [36], and have achieved rich theoretical research results. However, few previous studies have taken homesickness as an object of study. This study introduces homesickness advertising to the field of brand sustainability, in order to provide intrinsic explanatory mechanisms for the impact of homesickness advertising on the hometown brand citizenship behaviors of consumers. Specifically, homesickness advertising can elicit positive brand behavior in consumers. This result is consistent with the findings of Zhang [13]. This study bridges the interdisciplinary field of research by linking psychological ownership in social psychology to brand citizenship behavior in the field of marketing, thus enriching the theoretical research related to local brands, to a certain extent, enhancing the localized connotation and promoting the sustainable development of brands.

Second, we applied psychological ownership to deconstruct the mechanism of consumer behavior toward hometown brands from the perspective of homesickness specific to China. Questionnaire data and contextual experimental research methods were used to explore the factors influencing consumer behaviors regarding hometown brand citizenship. Based on the characteristics of hometown brands, an experimental research method was used to explore the mechanism of the effect of homesickness advertising on the of hometown brand citizenship behavior of consumers. We explored the emotional pathways of consumer civic behavior formation and enhancement in the homesickness advertising context. Expanding the theoretical margins of psychological ownership, place attachment, and consumer brand citizenship behavior is important for promoting sustainable brand development.

### 4.3. Research Insights

Firstly, modern society is characterized by frequent population movements and a significant increase in the number of migrant workers. Therefore, companies can emphasize the hometown message and emotional experience behind the brand in their brand marketing planning, effectively using and expanding the product source and hometown sentiment effect so that consumers of that brand can extract the stability factor from it, thus generating citizenship behavior towards that brand and promoting sustainable brand development.

Second, companies can take advantage of the mechanisms inherent in homesickness advertising to promote citizenship behavior toward hometown brands. By emphasizing the connection between a product, the brand, and the consumer’s hometown, the consumer extracts a stable message about the hometown. This stimulates psychological ownership of the hometown brand in consumers, thus building broader place attachment and positively promoting their civic behavior towards the hometown brand.

### 4.4. Limitations and Future Research

In this paper, we mainly used experimental methods for our research. In future research, field experiments can be carried out in other countries, and the time interval between experiments can be expanded in order to further improve the external validity. First, we conducted manipulation experiments through the reading of advertising materials in the experiment. Although the manipulation effect could be achieved, we used virtual brands and only studied Chinese consumer behaviors. Second, in addition to brand citizenship behavior, other consumer psychological and behavioral factors may also be influenced by nostalgia-based advertisements. Therefore, future research can target advertisements focused on real consumer hometown brands. The impact of nostalgia-based advertising on consumer brand behavior may be further observed in real consumer environments, such as shopping malls and shopping websites. The results of such studies will make the findings of this article more authentic and practical which, in turn, will improve its external validity.

## Figures and Tables

**Figure 1 behavsci-13-00054-f001:**
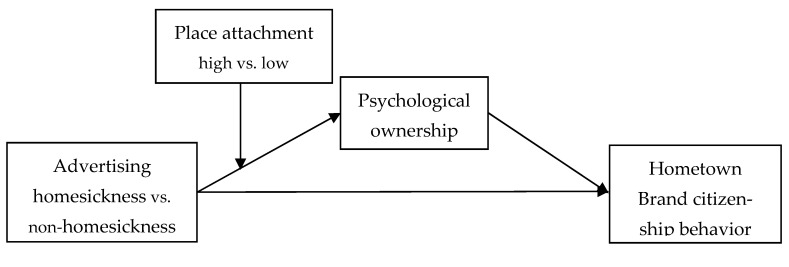
Study model.

**Figure 2 behavsci-13-00054-f002:**
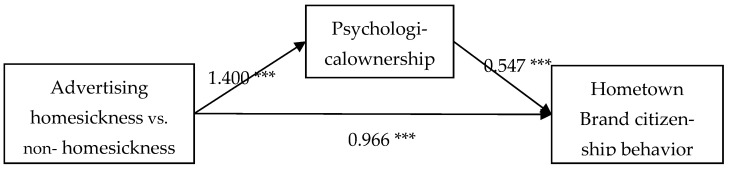
Mediated effect model test of the influence of homesickness advertising (homesickness vs. non- homesickness) on hometown brand citizenship behavior of consumers. Note: *** *p* < 0.001 (bilateral).

**Table 1 behavsci-13-00054-t001:** Independent samples *t*-test for homesickness advertisements (vs. non-homesickness advertisements).

	Ad Type	Number of Cases	M	SD	SE	*t*	*p*
Hometown Brand Citizenship Behavior	Homesickness advertising	77	5.243	1.113	0.127	2.190	**
	Non-homesickness advertising	77	4.866	1.018	0.116		

Note: ** *p* < 0.01 (bilateral).

**Table 2 behavsci-13-00054-t002:** Independent samples *t*-test for psychological ownership.

	Ad Type	Number of Cases	M	SD	SE	*t*	*p*
Hometown Brand Citizenship Behavior	Homesickness advertising	77	5.405	0.892	0.099	−7.500	***
	Non-homesickness advertising	77	4.005	1.412	0.0158		

Note: *** *p* < 0.001 (bilateral).

**Table 3 behavsci-13-00054-t003:** Mediated effect model test of the influence of homesickness advertising (homesickness vs. non-homesickness) on hometown brand citizenship behavior of consumers.

Variables	Hometown Brand Citizenship Behavior		Psychological Ownership		Hometown Brand Citizenship Behavior	
	b	*t*	b	*t*	b	*t*
Constant	4.569 ***	40.644	4.005 ***	30.341	2.376 ***	0.226
IV						
advertising (homesickness vs. non-homesickness)	0.966 ***	6.074	1.400 ***	7.499	0.199	0.142
Mediating variable						
psychological ownership					0.547 ***	10.514
R^2^	0.189		0.263		0.524	
F(df)	36.896 ***		56.245 ***		86.517 ***	

Note: *** *p* < 0.001 (bilateral).

**Table 4 behavsci-13-00054-t004:** Comparative analysis of direct and indirect effects.

	Mediator Variable	Effect	SE	*t*	95%		Relative Effect
					LLCI	ULCI	
Total effect		0. 965	0.159	6.074	0.651	1.279	
Direct effect		0.199	0.142	1.400	−0.082	0.480	20.6%
Indirect effect	Psychological ownership	0.766	0.134	-	0.524	1.046	79.8%

**Table 5 behavsci-13-00054-t005:** Results of the test for mediating effects (Bootstrap).

Moderated Variable	Mediated Variable	Moderated Level	Indirect Effect	SE	95% CI	
					LLCI	ULCI
Place attachment	Psychological ownership	High place attachment	0.952	0.129	0.710	1.214
		Low place attachment	0.582	0.116	0.372	0.832

## Data Availability

All data are available via email from the corresponding author.

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
