# Peer review of "The Hometown Is Hard to Leave, the Homesickness Is Unforgettable—The Influence of Homesickness Advertisement on Hometown Brand Citizenship Behavior of Consumers"

_behavsci, 2023, doi:10.3390/bs13010054_

Round 1
Reviewer 1 Report
The authors investigate the impact of homesickness advertising on consumers' hometown brand citizenship behavior. Specifically, the findings show that homesickness advertising can activate people's psychological ownership, which in turn increases consumers' hometown brand citizenship behavior. The topic is both theoretical and practical importance. But, I do have some concerns, roughly divided into conceptual and empirical.
Conceptual issues
1. In the first paragraph of the Introduction, the authors use a lot of space to describe what is homesickness, which is quite common sense. Therefore, I did not get the whole thing about what homesickness advertising is after reading the first paragraph. Since homesickness advertising instead of homesickness is the core construct of this research, I suggest the author get to the homesickness advertising directly with some examples to show the prevalence and importance of homesickness advertising.
2. The second paragraph of the Introduction mentions consumers' brand behavior. However, in the third paragraph, the authors change the concept to hometown brand citizenship behavior. What are the definitions of the two concepts? What are the conceptual differences between the two concepts?
3. Although the ideas are interesting, the paper in its current form is not making as much of a theoretical contribution as it potentially could. In particular, the Introduction should state the theoretical contributions made by the current research. However, in this version of the manuscript, the authors do not clarify why examining the relationship between homesickness advertising and consumers' hometown brand citizenship behavior is theoretically important. Furthermore, there are no citations from any previous literature. Authors should state the theoretical contribution based on existing literature.
4. The definition of some constructs in the current research is vague. For example, the definition of consumer brand citizenship behavior is adapted from citizenship behavior from organizational behavior research. But the authors fail to define consumer brand citizenship behavior clearly and clarify the conceptual differences between consumer brand citizenship behavior and citizenship behavior from organizational behavior research.
5. I suggest the authors to combine 2.1.2. and 2.1.4. together as they are related to consumer brand citizenship behavior.
6. The moderating variable is place attachment. However, the manuscript lacks a literature review on place attachment. In particular, the definition of place attachment is unclear. Although the authors cite research on emotional attachment (lines 253-286), how emotional attachment and place attachment are related to each other?
Empirical issues
1. There are several pre-experiments. For example, 3.1. and 3.2.1. are both used to test the effectiveness of manipulation materials. However, since 3.1. has already tested the manipulation effectiveness, why do the authors conduct another pre-experiment test (i.e., 3.2.1.)?
2. The measures used to do the manipulation check are biased. The subjects were asked to respond to the question "What is the type of the above advertisement (1="homesickness advertising, " 7=" non-homesickness advertising"). However, do the subjects understand the definition of homesickness advertising? In other words, do subjects differentiate clearly between homesickness advertising and non-homesickness advertising? Instead, asking subjects to rate the extent they miss their hometown after watching the advertisement seems to be a better measure.
3. The manipulation materials are not clear enough. For example, in experiment one, subjects in the homesickness advertising condition watch an advertisement with a Chinese-New-Year theme. Although people's homesickness might be activated by the Chinese New Year theme, they can also be reminded of the festival atmosphere and positive emotions by watching a holiday-themed advertisement. However, the authors fail to measure these confounding variables that could bias the results.
4. Authors might argue that subjects are informed that the product used in the advertisement is from their hometown, which can be effective in priming homesickness. However, it is the product that activates homesickness not the advertisement per se. In other words, the materials may manipulate the homesickness products instead of homesickness advertisements.
5. One of the items (i.e., "If people around need to buy similar products, I would recommend them to buy the brand advertised above in preference to other hometown brands") seems to measure people's recommendation tendency instead of brand citizenship behavior. The authors should further clarify why this item present brand citizenship behavior.
Minor issues
1. There are several typos and grammar issues in the manuscript. For example, "Homesickness has some homesickness meaning, has 36 gradually become a social and cultural psychology" (lines 36-37).
2. I highly suggest the authors to find a copy editor to assist you with any grammar and language issues.
Author Response
RESPONSES TO REVIEWER # 1
I am very glad to receive your suggestions and thank you for your hard work, they have helped me a lot to improve the manuscript. In the meantime, I have uploaded the revised manuscript in the attached widget with the changes highlighted in blue in the new manuscript. Below are specific notes about my revisions.
Conceptual issues
Comment 1
In the first paragraph of the Introduction, the authors use a lot of space to describe what is homesickness, which is quite common sense. Therefore, I did not get the whole thing about what homesickness advertising is after reading the first paragraph. Since homesickness advertising instead of homesickness is the core construct of this research, I suggest the author get to the homesickness advertising directly with some examples to show the prevalence and importance of homesickness advertising.
Authors’ response
Thank you so much for your input! I very much agree with your opinion, it is very professional and has been of great help to my manuscript revision. In response to your comments, I have reorganized the descriptions of nostalgic advertisements in the "Introduction" and "Literature Review" sections, and added relevant examples to illustrate the prevalence and importance of homesick advertisements, and updated the references References to the literature are as follows:
- Introduction
In the current context of COVID-19 prevention and control, the new epidemic is still playing out, with ups and downs and small outbreaks all the time, making it increasingly difficult for people to return home every holiday season. In our modern society, with frequent population movements and increased pressure of life, the outbreak of homesickness is an inevitable stage of social reform and development. People who are far away are haunted by the memories of their hometowns, and people under the high pressure of urban life also yearn for the traditional rural life, in order to return to the basics. Homesickness has a certain meaning, and has gradually become a social and cultural psychological phenomenon. There have been numerous classic advertising works related to homesickness. The various industries that play the homesickness card, whether it is agritourism, rural tourism, various consumer goods labeled to promote homesickness, or even homesickness advertising, entertainment programs, and so on, have been recognized and accepted by the public [1]. The most direct and common expression of homesickness theme in advertising is that people who are far away from their hometown miss their hometown and family during various festivals. From the early "Confucian house wine, making people homesick", to the "Spring Festival Home" series of public service advertisements launched by CCTV during the Chinese New Year in recent years, such campaigns are all affectionate interpretations of homesickness advertising. At a time when remembering homesickness is strongly advocated, the hometown brand—as a symbol representing a specific region—can induce homesickness feelings in the hearts of consumers. Especially for those who live in other countries, adding the element of homesickness into hometown brand advertising can better stimulate an inner sense of belonging and self-identity. This will improve the hometown brand citizenship behavior of consumers and promote sustainable brand development.
- Literature Review and Theoretical Hypotheses
2.1. Literature Review
2.1.1. Homesickness advertising
Homesickness advertising tends to focus on the homesickness memories of different consumer subjects. Homesickness can also be expressed in terms of folkloric events. To some extent, homesickness has developed accordingly with modernity. Liu has stated that, in advertising, the most common theme of homesickness is the longing for hometown and relatives, which is strongly reflected in the homesickness of people who are far away from home and work in different places. To a certain extent, this conforms to the interpretation of the meaning of "homesickness", and also caters to the homesickness psychology of some advertising audiences [1]. He has analyzed 3 data sets including 4091 words from B station, revealing that some CCTV Spring Festival Gala PSAs have been broadcast for many years, but still continue to attract views and discussion on video sites [8]. Netizens have expressed their memories of the commercial and their emotions for their hometown through pop-ups. This also indicates that the CCTV Spring Festival Gala PSAs can invoke emotional resonance in viewers and summon up people's homesickness memories. Homesickness for one’s hometown is a permanent topic for Chinese people. Triggering emotional resonance in the advertising audience through the emotion of homesickness is an important path for traditional culture, in order to improve the psychological fit of audiences in advertising media. The creation of homesickness and re-interpretation of folklore imagery cater to the homesickness mentality of the audience, mobilizing people's memories of the past, providing a convenient means for the development of folklore resources, and helping to promote the regeneration and recognition of traditional folklore [9].
Comment 2
The second paragraph of the Introduction mentions consumers' brand behavior. However, in the third paragraph, the authors change the concept to hometown brand citizenship behavior. What are the definitions of the two concepts? What are the conceptual differences between the two concepts?
Authors’ response
Thank you very much for your improvement suggestions on the shortcomings of my thesis. Regarding the difference between these two concepts, I will combine your suggestions for revision in the fifth question, and in 1 and 2.1.2, the relationship between brand citizenship behavior and hometown brand The definition of civic behavior has been differentiated and related concepts have been rearranged and revised, as follows:
- Introduction
According to previous studies, the view that consumers are "part-time employees" or "partial employees" of enterprises has been recognized by many scholars [2]. GROTH has formally proposed the concept of "customer citizenship behaviors (CCB)," corresponding to the "part-time employee" status of consumers, on the basis of organizational citizenship behavior-related research. and defined it as a service that is not necessary for the successful production or delivery of a service. However, the spontaneous, haphazard behavior of consumers is generally good for the brand as a whole [3]. Hometown brands are those that originate from a consumer's hometown. The civic behavior of consumers towards their hometown brands can help to reduce the operating costs of these brands. This allows the hometown brand to obtain more valuable business information and enhance their competitive advantage. The existing literature has mostly examined consumer brand behavior from the perspective of product functionality and rational cognition, which is considered to be weak [4,7]. Can homesickness advertising have an impact on consumers living in other countries, leading to a series of brand citizenship behaviors towards their home brands? What are the variables by which the mechanism of this role is realized? Can hometown branding play a role in the emotional content of the brand to promote sustainable development? These are the questions that must be answered, if such brands are to be further developed and broadened.
2.1.2. Consumer Hometown Brand Citizenship Behavior
The concept of consumer brand citizenship behavior originates from the research field of Organizational Citizenship Behavior. Gruen was the first to apply the concept of "citizen behavior" in organizational behavior to research on consumer behavior in marketing [10]. Groth has formally defined consumer brand citizenship behavior as follows: consumers voluntarily adopt products or services that benefit brands. However, it is not a necessary measure for brands to provide products and services. Voluntary initiative, positivity, and non-role are important characteristics of consumer brand citizenship behavior [3]. Chen has defined consumer brand citizenship behavior as a spontaneous non-consumption behavior or service production behavior outside the consumer's role, which is generally beneficial to the brand or other consumers [11]. Consumer hometown brand citizenship behavior is a new concept in the field of consumer behavior research, which means that consumers voluntarily present valuable and constructive behaviors for hometown brands outside their roles. This can lead to low-cost (or even no-cost) competitive advantages for hometown enterprises. For example, consumers spontaneously recommend, through word-of-mouth behavior, their hometown brand to their friends; are willing to show that they own the products or services of the hometown brand for publicity; actively cooperate with the brand's return visit research activities or new product development activities; and are willing to cooperate with the brand wait. Li has stated that regional brands clearly and specifically inform the geographical origin of products [12]. Therefore, they can activate the sense of self-identity in consumers from the same region, providing important homesickness sustenance for people who are far away from their hometown and miss their hometown in the modern context of rapid urbanization and modernization. Zhang has pointed out that consumers are also more inclined to accept marketing strategies that conform to their self-identity. Therefore, people tend to support brands from their hometown in order to gain an emotional sense of belonging. When a person’s homesickness is relatively high, they are more likely to transfer their homesickness and love for their hometown to their hometown brands [13].
Comment 3
Although the ideas are interesting, the paper in its current form is not making as much of a theoretical contribution as it potentially could. In particular, the Introduction should state the theoretical contributions made by the current research. However, in this version of the manuscript, the authors do not clarify why examining the relationship between homesickness advertising and consumers' hometown brand citizenship behavior is theoretically important. Furthermore, there are no citations from any previous literature. Authors should state the theoretical contribution based on existing literature.
Authors’ response
Your suggestions are of great help to me, thank you again for your valuable comments, regarding the revision of this part, in the introduction part I added the theoretical contribution made by the current research, and added supplementary nostalgia advertisements and The relationship between civic behaviors, with additional references to the literature, is as follows:
- Introduction
According to previous studies, the view that consumers are "part-time employees" or "partial employees" of enterprises has been recognized by many scholars [2]. GROTH has formally proposed the concept of "customer citizenship behaviors (CCB)," corresponding to the "part-time employee" status of consumers, on the basis of organizational citizenship behavior-related research. and defined it as a service that is not necessary for the successful production or delivery of a service. However, the spontaneous, haphazard behavior of consumers is generally good for the brand as a whole [3]. Hometown brands are those that originate from a consumer's hometown. The civic behavior of consumers towards their hometown brands can help to reduce the operating costs of these brands. This allows the hometown brand to obtain more valuable business information and enhance their competitive advantage. The existing literature has mostly examined consumer brand behavior from the perspective of product functionality and rational cognition, which is considered to be weak [4,7]. Can homesickness advertising have an impact on consumers living in other countries, leading to a series of brand citizenship behaviors towards their home brands? What are the variables by which the mechanism of this role is realized? Can hometown branding play a role in the emotional content of the brand to promote sustainable development? These are the questions that must be answered, if such brands are to be further developed and broadened.
Using psychological ownership as a mediating variable and place attachment as a moderating variable, we investigate the effect of homesickness advertising on the hometown brand citizenship behavior of consumers, with the following main contributions:
First, we enrich the theoretical research related to brand citizenship behavior, thus providing a strong basis for sustainable development of consumer hometown brands. This study explores the relationships between homesickness advertising, psychological ownership, place attachment, and consumer brand citizenship behavior. The results suggest that consumer hometown brands can deliver value through consumer brand citizenship behaviors, and can play a role in promoting positive consumer purchases and even spreading positive word-of-mouth. This serves to promote the sustainable development of the branding path of hometown products.
Second, we provide a basis and guidance for corporate branding marketing, by exploring effective countermeasures for consumer brand behavior enhancement from the perspective of consumer psychological ownership. This allows brand production operators and relevant government departments to dig deep and cultivate such brands, by providing a theoretical basis and practical guidance to promote sustainable brand development
The remainder of this paper is organized as follows: The theoretical background, conceptualization, and hypothesis development are presented first. Next, the research methodology, including the research design, study materials, and findings, is presented. The final section summarizes the analysis of the experimental findings, and suggests management insights to promote sustainable brand development.
References
- Mills. P. K, Morris. J. H. Clients as “Partial” Employees of Service Organizations: Role Development in Client Participation. Academy of Management Review,1986,11(4), 726-735. doi: 10.5465/amr.1986.4283916.
- Groth. M. Customers as Good Soldiers:Examining Citizenship Behaviors in Internet Service Deliveries. Journal of Management,2005,31(1):7-27. DOI:10.1177/0149206304271375.
- Liu, H.; Yang, J.; Chen, X. Making the Customer-Brand Relationship Sustainable: The Different Effects of Psychological Contract Breach Types on Customer Citizenship Behaviours. Sustainability 2020, 12, 630. doi:10.3390/su12020630
- Ezgi Erkmen. All Consumers Are Same for the Effect of Brand Citizenship Behaviors?: The Role of Nationality. Journal of International Journal of Marketing Studies, 2014,Vol. 6, No. 3.pp:65-75. doi 10.5539/ijms.v6n3p65
- Nicoleta DOSPINESCU. "The Public Relations Events in Promoting Brand Identity of the City," Economics and Applied Informatics, "Dunarea de Jos" University of Galati, Faculty of Economics and Business Administration, , 2014, issue 1, 39-46.
- Helm, S.V., Renk, U. and Mishra, A., "Exploring the impact of employees’ self-concept, brand identification and brand pride on brand citizenship behaviors", European Journal of Marketing, 2016,Vol. 50 No. 1/2, 58-77. doi:10.1108/EJM-03-2014-0162
Comment 4
The definition of some constructs in the current research is vague. For example, the definition of consumer brand citizenship behavior is adapted from citizenship behavior from organizational behavior research. But the authors fail to define consumer brand citizenship behavior clearly and clarify the conceptual differences between consumer brand citizenship behavior and citizenship behavior from organizational behavior research.
Authors’ response
I am very grateful for your comments on the fuzzy definition of related concepts in the thesis. Your comments are of great help to my thesis. First of all, in the "Literature Review" section, I have reorganized the definition of consumer brand citizenship behavior, and Redefining the definition of citizenship behavior and brand citizenship behavior from organizational behavior, I have made the following revisions to your comments:
- Introduction
According to previous studies, the view that consumers are "part-time employees" or "partial employees" of enterprises has been recognized by many scholars [2]. GROTH has formally proposed the concept of "customer citizenship behaviors (CCB)," corresponding to the "part-time employee" status of consumers, on the basis of organizational citizenship behavior-related research. and defined it as a service that is not necessary for the successful production or delivery of a service. However, the spontaneous, haphazard behavior of consumers is generally good for the brand as a whole [3]. Hometown brands are those that originate from a consumer's hometown. The civic behavior of consumers towards their hometown brands can help to reduce the operating costs of these brands. This allows the hometown brand to obtain more valuable business information and enhance their competitive advantage. The existing literature has mostly examined consumer brand behavior from the perspective of product functionality and rational cognition, which is considered to be weak [4,7]. Can homesickness advertising have an impact on consumers living in other countries, leading to a series of brand citizenship behaviors towards their home brands? What are the variables by which the mechanism of this role is realized? Can hometown branding play a role in the emotional content of the brand to promote sustainable development? These are the questions that must be answered, if such brands are to be further developed and broadened.
2.1.2. Consumer Hometown Brand Citizenship Behavior
The concept of consumer brand citizenship behavior originates from the research field of Organizational Citizenship Behavior. Gruen was the first to apply the concept of "citizen behavior" in organizational behavior to research on consumer behavior in marketing [10]. Groth has formally defined consumer brand citizenship behavior as follows: consumers voluntarily adopt products or services that benefit brands. However, it is not a necessary measure for brands to provide products and services. Voluntary initiative, positivity, and non-role are important characteristics of consumer brand citizenship behavior [3]. Chen has defined consumer brand citizenship behavior as a spontaneous non-consumption behavior or service production behavior outside the consumer's role, which is generally beneficial to the brand or other consumers [11]. Consumer hometown brand citizenship behavior is a new concept in the field of consumer behavior research, which means that consumers voluntarily present valuable and constructive behaviors for hometown brands outside their roles. This can lead to low-cost (or even no-cost) competitive advantages for hometown enterprises. For example, consumers spontaneously recommend, through word-of-mouth behavior, their hometown brand to their friends; are willing to show that they own the products or services of the hometown brand for publicity; actively cooperate with the brand's return visit research activities or new product development activities; and are willing to cooperate with the brand wait. Li has stated that regional brands clearly and specifically inform the geographical origin of products [12]. Therefore, they can activate the sense of self-identity in consumers from the same region, providing important homesickness sustenance for people who are far away from their hometown and miss their hometown in the modern context of rapid urbanization and modernization. Zhang has pointed out that consumers are also more inclined to accept marketing strategies that conform to their self-identity. Therefore, people tend to support brands from their hometown in order to gain an emotional sense of belonging. When a person’s homesickness is relatively high, they are more likely to transfer their homesickness and love for their hometown to their hometown brands [13].
Comment 5
I suggest the authors to combine 2.1.2. and 2.1.4. together as they are related to consumer brand citizenship behavior.
Authors’ response
Thank you very much for pointing this out, I've done some work on combining 2.1.2 with 2.1.4. For your convenience, I have pasted the revised paragraph below.
2.1.2. Consumer Hometown Brand Citizenship Behavior
The concept of consumer brand citizenship behavior originates from the research field of Organizational Citizenship Behavior. Gruen was the first to apply the concept of "citizen behavior" in organizational behavior to research on consumer behavior in marketing [10]. Groth has formally defined consumer brand citizenship behavior as follows: consumers voluntarily adopt products or services that benefit brands. However, it is not a necessary measure for brands to provide products and services. Voluntary initiative, positivity, and non-role are important characteristics of consumer brand citizenship behavior [3]. Chen has defined consumer brand citizenship behavior as a spontaneous non-consumption behavior or service production behavior outside the consumer's role, which is generally beneficial to the brand or other consumers [11]. Consumer hometown brand citizenship behavior is a new concept in the field of consumer behavior research, which means that consumers voluntarily present valuable and constructive behaviors for hometown brands outside their roles. This can lead to low-cost (or even no-cost) competitive advantages for hometown enterprises. For example, consumers spontaneously recommend, through word-of-mouth behavior, their hometown brand to their friends; are willing to show that they own the products or services of the hometown brand for publicity; actively cooperate with the brand's return visit research activities or new product development activities; and are willing to cooperate with the brand wait. Li has stated that regional brands clearly and specifically inform the geographical origin of products [12]. Therefore, they can activate the sense of self-identity in consumers from the same region, providing important homesickness sustenance for people who are far away from their hometown and miss their hometown in the modern context of rapid urbanization and modernization. Zhang has pointed out that consumers are also more inclined to accept marketing strategies that conform to their self-identity. Therefore, people tend to support brands from their hometown in order to gain an emotional sense of belonging. When a person’s homesickness is relatively high, they are more likely to transfer their homesickness and love for their hometown to their hometown brands [13].
Comment 6
The moderating variable is place attachment. However, the manuscript lacks a literature review on place attachment. In particular, the definition of place attachment is unclear. Although the authors cite research on emotional attachment (lines 253-286), how emotional attachment and place attachment are related to each other?
Authors’ response
I very much agree with your suggestion, to which I added 2.1.4 Literature review of place attachment, as follows:
2.1.4. Place Attachment
Williams and Roggenbuck first proposed the concept of "Place Attachment;" that is, an individual's sense of belonging to a special place [19]. In the field of brand marketing, scholars have extended the attachment relationship in psychology from "people and people" to "people and things". The higher the degree of emotional association between the consumer and a place, the higher the consumer's preference for local brand products, resulting in loyal buying behavior, ultimately achieving corporate brand marketing goals. In terms of commonality, scholars believe that the emotional connection of consumers to places is an important part of place attachment, and have emphasized the important role of emotional investment. Takamatsu has stated that place attachment refers to a unique emotional bond between an individual and a place, and the strength of this relationship is important to whether the individual wants to spend more time, energy, and money on the place [20]. Stefaniak et al. have further defined it as a positive emotional connection between an individual and a specific place, with the main characteristic being that the individual tends to maintain a long-term and close connection with the place [21]. In terms of personality, the concept of place attachment by different scholars includes a personalized part, focused on bond relationship and emphasized feelings, among other aspects [22].
Empirical issues
Comment 1
There are several pre-experiments. For example, 3.1. and 3.2.1. are both used to test the effectiveness of manipulation materials. However, since 3.1. has already tested the manipulation effectiveness, why do the authors conduct another pre-experiment test (i.e., 3.2.1.)?
Authors’ response
Thank you for your valuable comments on the paper. Regarding your questions, our considerations are as follows: Since the experiment is not done at one time, and in order to consolidate the external validity of the experimental model, we have adopted different experiments in different experiments. Stimulus materials, so in order to ensure the effectiveness of each experimental material, we conducted a pre-experiment test before each experiment.
Comment 2
The measures used to do the manipulation check are biased. The subjects were asked to respond to the question "What is the type of the above advertisement (1="homesickness advertising, " 7=" non-homesickness advertising"). However, do the subjects understand the definition of homesickness advertising? In other words, do subjects differentiate clearly between homesickness advertising and non-homesickness advertising? Instead, asking subjects to rate the extent they miss their hometown after watching the advertisement seems to be a better measure.
Authors’ response
Thank you very much for your opinion. We have thought carefully about your opinion and made the following explanations: First of all, during the experimental operation, we explained the difference between homesickness advertisements and non-homesickness advertisings to the subjects when they answered the questions. definition, and its concept is further explained in the questionnaire to ensure that the subjects can further distinguish between homesickness advertisings and non-homesickness advertisings. Secondly, before each experiment, we conducted a pre-experiment, and the experimental results were the same It was shown that there was a significant difference between the experimental group and the control group, which further proved that the experimental material was effective.
At the same time, in order to further solve this problem, we added an in-group experiment, and the results still showed that the subjects could still better distinguish between homesickness advertisings and non-homesickness advertisings. If we simply discuss homesickness, it may be out of the marketing context. Thanks again for your valuable comments.
Comment 3
The manipulation materials are not clear enough. For example, in experiment one, subjects in the homesickness advertising condition watch an advertisement with a Chinese-New-Year theme. Although people's homesickness might be activated by the Chinese New Year theme, they can also be reminded of the festival atmosphere and positive emotions by watching a holiday-themed advertisement. However, the authors fail to measure these confounding variables that could bias the results.
Authors’ response
We've noticed this and appreciate your input. We conducted an additional experiment, mainly on the basis of the original experiment, to measure the level of their emotions, and included it as a covariate in the quantitative analysis. The results showed that their emotions had no significant impact on the results.
Incorporating the control variable emotion into the model as a covariate for the main effect analysis again, it was found that the main effect of nostalgia advertisements on consumer hometown brand citizenship behavior is still significant F(1, 60.483)=226.658, P=0.000, the influence of emotion Not significant, F(1, 0.631) 2.366=, P=0.129>0.1.
Comment 4
Authors might argue that subjects are informed that the product used in the advertisement is from their hometown, which can be effective in priming homesickness. However, it is the product that activates homesickness not the advertisement per se. In other words, the materials may manipulate the homesickness products instead of homesickness advertisements.
Authors’ response
Thank you for your suggestion. We have thought carefully about your question and made the following reply: In the experiment, the manipulation materials we used were all the same product and the same brand. The only difference between the experimental group and the control group is the type of advertisement, which is enough to show that the difference between the experimental group and the control group is the effect of the advertisement itself. And during each experimental operation, we replaced the manipulated materials for each experiment.
Comment 5
One of the items (i.e., "If people around need to buy similar products, I would recommend them to buy the brand advertised above in preference to other hometown brands") seems to measure people's recommendation tendency instead of brand citizenship behavior. The authors should further clarify why this item present brand citizenship behavior.
Authors’ response
Your opinion is very professional. Regarding your opinion, we have further clarified in the "Literature Review" and "Research Hypotheses" what items will cause consumers' hometown brand citizenship behavior. The specific content is as follows:
2.1.2. Consumer Hometown Brand Citizenship Behavior
The concept of consumer brand citizenship behavior originates from the research field of Organizational Citizenship Behavior. Gruen was the first to apply the concept of "citizen behavior" in organizational behavior to research on consumer behavior in marketing [10]. Groth has formally defined consumer brand citizenship behavior as follows: consumers voluntarily adopt products or services that benefit brands. However, it is not a necessary measure for brands to provide products and services. Voluntary initiative, positivity, and non-role are important characteristics of consumer brand citizenship behavior [3]. Chen has defined consumer brand citizenship behavior as a spontaneous non-consumption behavior or service production behavior outside the consumer's role, which is generally beneficial to the brand or other consumers [11]. Consumer hometown brand citizenship behavior is a new concept in the field of consumer behavior research, which means that consumers voluntarily present valuable and constructive behaviors for hometown brands outside their roles. This can lead to low-cost (or even no-cost) competitive advantages for hometown enterprises. For example, consumers spontaneously recommend, through word-of-mouth behavior, their hometown brand to their friends; are willing to show that they own the products or services of the hometown brand for publicity; actively cooperate with the brand's return visit research activities or new product development activities; and are willing to cooperate with the brand wait. Li has stated that regional brands clearly and specifically inform the geographical origin of products [12]. Therefore, they can activate the sense of self-identity in consumers from the same region, providing important homesickness sustenance for people who are far away from their hometown and miss their hometown in the modern context of rapid urbanization and modernization. Zhang has pointed out that consumers are also more inclined to accept marketing strategies that conform to their self-identity. Therefore, people tend to support brands from their hometown in order to gain an emotional sense of belonging. When a person’s homesickness is relatively high, they are more likely to transfer their homesickness and love for their hometown to their hometown brands [13].
2.2.2. Homesickness advertising (vs. non-homesickness advertising) and consumer hometown brand citizenship behavior: the mediating role of psychological ownership
.....According to social exchange theory, consumers who develop psychological ownership of their hometown brand will enhance their brand behavior, as a result. This is demonstrated by buying more, paying a premium, word-of-mouth referrals, and so on, as well as always being ready to maintain the brand. In the stimulus—individual physiological/psychological response pathway, utilization of the homesickness advertising context induces and changes the emotional experience of consumers. This affects emotional changes in consumers and creates psychological ownership of the hometown brand. After a series of physiological and psychological reaction process, consumers will enact a behavioral response to the stimulus, such as re-purchasing, recommendation, and other convergent behavior brand citizenship behavior.
Minor issues
Comment 1
There are several typos and grammar issues in the manuscript. For example, "Homesickness has some homesickness meaning, has 36 gradually become a social and cultural psychology" (lines 36-37).
Authors’ response
I am very sorry to see this problem, because after checking, I confirmed that there is a writing error. Thank you again for pointing this out, and I am ashamed of my lack of rigor. I have revised it and enlisted the help of a professional English speaker.
Comment 2
I highly suggest the authors to find a copy editor to assist you with any grammar and language issues.
Authors’ response
Thank you very much for your suggestion, I have submitted my article to MDPI professional English editor for English revision, thank you again for your help on my article!

Reviewer 2 Report
Dear Authors,
Thank you for the opportunity to review the article entitled "It's hard to leave my hometown behind:The Influence of Homesickness Advertisement on Consumers ' Hometown Brand Citizenship Behavior".
Below you can find my recommendations.
First of all, please avoid using verbs ("It's hard...") in the title of a scientific article. A title should be impersonal and formal.
In the Introducion, at rows 37-38, you have a sequence of text without verb: "From the social field in the 19th 37 century to the habitat field in the 20th century."
Please revise this sequence so that it becomes a sentence.
English should be revised. For example, at rows 48-49 you have an "a" in the middle of the sentence, after the word "homesickness": "At a time when remembering homesickness a is strongly advocated, the hometown brand, as a symbol representing a specific region."
Please correct it.
At rows 61-62 you say that "This is the question that must be answered if the brand can be further developed and broadened.", but right before this text, you have no less than three (3) questions.
I recommend you to define within the Introduction the following aspects: the research gap, the research goal and the research question.
The title of the section "2.1.1. homesickness advertising" should start with capital letter.
The same remark for "2.2.1. homesickness advertising (vs. non- homesickness advertising) and consumer hometown brand citizenship behavior".
The section "2.1.2. Consumer Brand Citizenship Behavior" must be improved. Here I recommend you to read and cite the following resources: https://doi.org/10.3390/su12020630, http://dx.doi.org/10.5539/ijms.v6n3p65, https://ideas.repec.org/a/ddj/fseeai/y2014i1p39-46.html, https://doi.org/10.1108/EJM-03-2014-0162. These references will define the concept of consumer brand citizenship behavior in the context of your research.
At the rows 364-365 you say that "The pre-experiment was conducted with a random sample of 30 subjects (14 female, age range 20-50 years) through Credamo's sample recommendation service." Normally, you should also specify the number of the male respondents (in this case, 16).
In the Conclusions chapter, I recommend you to add a section to present your research limitations. Think about:
- the regional limitations (the study was conducted in China);
- the reduced size of the sample;
- the time interval when the research was conducted.
Dear Authors,
I really hope my recommendations will be useful for you in order to improve the paper.
Best Regards!
Author Response
RESPONSES TO REVIEWER # 2
I am very glad to receive your suggestions and thank you for your hard work, they have helped me a lot to improve the manuscript. In the meantime, I have uploaded the revised manuscript in the attached widget with the changes highlighted in blue in the new manuscript. Below are specific notes about my revisions.
Comment 1
First of all, please avoid using verbs ("It's hard...") in the title of a scientific article. A title should be impersonal and formal.
Authors’ response
Thank you very much for your valuable comments on my article. I recognize your opinion very much. Your opinion is of great help to me. Regarding your opinion, I have modified the title of the article as follows:
The Hometown is Hard to Leave, The Homesickness is Unforgettable—The Influence of Homesickness Advertisement on Hometown Brand Citizenship Behavior of Consumers
Comment 2
In the Introducion, at rows 37-38, you have a sequence of text without verb: "From the social field in the 19th 37 century to the habitat field in the 20th century."
Please revise this sequence so that it becomes a sentence.
Authors’ response
I am very sorry to see this problem, because after checking, I confirmed that there is a writing error. Thank you again for pointing this out, and I am ashamed of my lack of rigor. And I have submitted my article to MDPI professional English editor for English revision, thank you again for your help on my article! The proof of the English editor is as follows:
Comment 3
English should be revised. For example, at rows 48-49 you have an "a" in the middle of the sentence, after the word "homesickness": "At a time when remembering homesickness a is strongly advocated, the hometown brand, as a symbol representing a specific region."
Please correct it.
Authors’ response
Your suggestion was very professional, we have made the following revision to this sentence, thank you again for your suggestion.
At a time when remembering homesickness is strongly advocated, the hometown brand—as a symbol representing a specific region—can induce homesickness feelings in the hearts of consumers.
Comment 4
At rows 61-62 you say that "This is the question that must be answered if the brand can be further developed and broadened.", but right before this text, you have no less than three (3) questions.
I am very sorry to see this problem, because after checking, I confirmed that there is a writing error and I wrote "These are" as "This is". Thank you again for pointing this out, and I am ashamed of my lack of rigor. I have corrected this part and pasted it below.
The existing literature has mostly examined consumer brand behavior from the perspective of product functionality and rational cognition, which is considered to be weak [4,7]. Can homesickness advertising have an impact on consumers living in other countries, leading to a series of brand citizenship behaviors towards their home brands? What are the variables by which the mechanism of this role is realized? Can hometown branding play a role in the emotional content of the brand to promote sustainable development? These are the questions that must be answered, if such brands are to be further developed and broadened.
Comment 5
I recommend you to define within the Introduction the following aspects: the research gap, the research goal and the research question.
Authors’ response
Your opinion is very professional, and your opinion has greatly helped me to revise my article, thank you again! Regarding this part, we have made the following changes:
According to previous studies, the view that consumers are "part-time employees" or "partial employees" of enterprises has been recognized by many scholars [2]. GROTH has formally proposed the concept of "customer citizenship behaviors (CCB)," corresponding to the "part-time employee" status of consumers, on the basis of organizational citizenship behavior-related research. and defined it as a service that is not necessary for the successful production or delivery of a service. However, the spontaneous, haphazard behavior of consumers is generally good for the brand as a whole [3]. Hometown brands are those that originate from a consumer's hometown. The civic behavior of consumers towards their hometown brands can help to reduce the operating costs of these brands. This allows the hometown brand to obtain more valuable business information and enhance their competitive advantage. The existing literature has mostly examined consumer brand behavior from the perspective of product functionality and rational cognition, which is considered to be weak [4,7]. Can homesickness advertising have an impact on consumers living in other countries, leading to a series of brand citizenship behaviors towards their home brands? What are the variables by which the mechanism of this role is realized? Can hometown branding play a role in the emotional content of the brand to promote sustainable development? These are the questions that must be answered, if such brands are to be further developed and broadened.
Using psychological ownership as a mediating variable and place attachment as a moderating variable, we investigate the effect of homesickness advertising on the hometown brand citizenship behavior of consumers, with the following main contributions:
First, we enrich the theoretical research related to brand citizenship behavior, thus providing a strong basis for sustainable development of consumer hometown brands. This study explores the relationships between homesickness advertising, psychological ownership, place attachment, and consumer brand citizenship behavior. The results suggest that consumer hometown brands can deliver value through consumer brand citizenship behaviors, and can play a role in promoting positive consumer purchases and even spreading positive word-of-mouth. This serves to promote the sustainable development of the branding path of hometown products.
Second, we provide a basis and guidance for corporate branding marketing, by exploring effective countermeasures for consumer brand behavior enhancement from the perspective of consumer psychological ownership. This allows brand production operators and relevant government departments to dig deep and cultivate such brands, by providing a theoretical basis and practical guidance to promote sustainable brand development
The remainder of this paper is organized as follows: The theoretical background, conceptualization, and hypothesis development are presented first. Next, the research methodology, including the research design, study materials, and findings, is presented. The final section summarizes the analysis of the experimental findings, and suggests management insights to promote sustainable brand development.
Comment 6
The title of the section "2.1.1. homesickness advertising" should start with capital letter.
The same remark for "2.2.1. homesickness advertising (vs. non- homesickness advertising) and consumer hometown brand citizenship behavior".
Authors’ response
Thank you very much for your suggestions for improvement on the deficiencies in my thesis, and I will make the following revisions:
2.2.1. Homesickness advertising (vs. non-homesickness advertising) and hometown brand citizenship behavior
Comment 7
The section "2.1.2. Consumer Brand Citizenship Behavior" must be improved. Here I recommend you to read and cite the following resources:
https://doi.org/10.3390/su12020630,
http://dx.doi.org/10.5539/ijms.v6n3p65,
https://ideas.repec.org/a/ddj/fseeai/y2014i1p39-46.html,
https://doi.org/10.1108/EJM-03-2014-0162.
These references will define the concept of consumer brand citizenship behavior in the context of your research.
Authors’ response
Your comments have been very professional and have helped me a lot in my articles, which I have included in my references after reading them carefully:
References
- Liu, H.; Yang, J.; Chen, X. Making the Customer-Brand Relationship Sustainable: The Different Effects of Psychological Contract Breach Types on Customer Citizenship Behaviours. Sustainability 2020, 12, 630. doi:10.3390/su12020630
- Ezgi Erkmen. All Consumers Are Same for the Effect of Brand Citizenship Behaviors?: The Role of Nationality. Journal of International Journal of Marketing Studies, 2014,Vol. 6, No. 3.pp:65-75. doi 10.5539/ijms.v6n3p65
- Nicoleta DOSPINESCU. "The Public Relations Events in Promoting Brand Identity of the City," Economics and Applied Informatics, "Dunarea de Jos" University of Galati, Faculty of Economics and Business Administration, , 2014, issue 1, 39-46.
- Helm, S.V., Renk, U. and Mishra, A., "Exploring the impact of employees’ self-concept, brand identification and brand pride on brand citizenship behaviors", European Journal of Marketing, 2016,Vol. 50 No. 1/2, 58-77. doi:10.1108/EJM-03-2014-0162
Comment 8
At the rows 364-365 you say that "The pre-experiment was conducted with a random sample of 30 subjects (14 female, age range 20-50 years) through Credamo's sample recommendation service." Normally, you should also specify the number of the male respondents (in this case, 16).
Authors’ response
Thank you very much for your comments, I have put the number of men interviewed into the experimental expression and marked it in blue in the article.Thank you very much for your input, I have included the number of men interviewed in the experimental statement.
Comment 9
In the Conclusions chapter, I recommend you to add a section to present your research limitations. Think about:
- the regional limitations (the study was conducted in China);
- the reduced size of the sample;
- the time interval when the research was conducted.
Authors’ response
Thank you very much for your professional revision opinion. I agree with your suggestion very much and actively revise it. The details are as follows:
4.4. Research shortcomings and prospects
In this paper, we mainly used experimental methods for our research. In future research, field experiments can be carried out in other countries, the time interval between experiments can be expanded in order to further improve the external validity. First, we conducted manipulation experiments through the reading of advertising materials in the experiment. Although the manipulation effect could be achieved, we used virtual brands and only studied Chinese consumer behaviors. Second, in addition to brand citizenship behavior, other consumer psychological and behavioral factors may also be influenced by nostalgia-based advertisements. Therefore, future research can target advertisements focused on real consumer hometown brands. The impact of nostalgia-based advertising on consumer brand behavior may be further observed in real consumer environments, such as shopping malls and shopping websites. The results of such studies will make the findings of this article more authentic and practical which, in turn, will improve its external validity.
Thanks again for your dedication and advice, and I wish you all the best!

Round 2
Reviewer 1 Report
The authors resolved major concerns in the first round. The current version meets the standard for acceptance.
Author Response
I am very grateful for your advice on my article, and very grateful for your hard work, which provides a great help to improve my article, thank you again.
Reviewer 2 Report
Dear Authors,
I have read the revised version of the manuscript and I appreciate your effort to improve the article. I consider that you addressed all my recommendations from the previous round of review.
In the present stage of the review process, I have only one minor recommendation: in the chapter "Conclusions and Recommendations" please also include a small paragraph containing the description of the limitations of your research.
Best Regards!
Author Response
RESPONSES TO REVIEWER # 2
I am very glad to receive your suggestions and thank you for your hard work, they have helped me a lot to improve the manuscript. In the meantime, I have uploaded the revised manuscript in the attached widget. Below are specific notes about my revisions.
Comment 1
In the present stage of the review process, I have only one minor recommendation: in the chapter "Conclusions and Recommendations" please also include a small paragraph containing the description of the limitations of your research.
Authors’ response
Thank you so much for your input! I very much agree with your opinion, it is very professional and has been of great help to my manuscript revision. In the chapter "Conclusions and Recommendations", we have revised the title of "4.4.limitations and future research" and explained the limitations of our article in this part. In order to better illustrate, I will modify this part Paste below:
4.4 Limitations and future research
In this paper, we mainly used experimental methods for our research. In future research, field experiments can be carried out in other countries, the time interval between experiments can be expanded in order to further improve the external validity. First, we conducted manipulation experiments through the reading of advertising materials in the experiment. Although the manipulation effect could be achieved, we used virtual brands and only studied Chinese consumer behaviors. Second, in addition to brand citizenship behavior, other consumer psychological and behavioral factors may also be influenced by nostalgia-based advertisements. Therefore, future research can target advertisements focused on real consumer hometown brands. The impact of nostalgia-based advertising on consumer brand behavior may be further observed in real consumer environments, such as shopping malls and shopping websites. The results of such studies will make the findings of this article more authentic and practical which, in turn, will improve its external validity.
Thanks again for your dedication and advice, and I wish you all the best!

Round 3
Reviewer 1 Report
The authors resolved the major concerns in this round. This version meets the standards for acceptance.
Author Response

(The authors gave the same response as above.)

Reviewer 2 Report
Dear Authors,
The article meets all my requirements from previous review rounds.
Best Regards!
Author Response

(The authors gave the same response as above.)
